# Fear of positive evaluation in borderline personality disorder

Anna Weinbrecht[1]*, Stefan Roepke[2], Babette Renneberg[1]

**1** Department of Clinical Psychology and Psychotherapy, Freie Universität Berlin, Berlin, Germany,
**2** Department of Psychiatry and Psychotherapy, Charité –Universitätsmedizin Berlin, corporate member of Freie Universität Berlin, Humboldt-Universität zu Berlin, and Berlin Institute of Health, Campus Benjamin Franklin, Berlin, Germany

* a.weinbrecht@fu-berlin.de

**Data Availability Statement:** All relevant data are within the manuscript and its Supporting Information files. We do not provide specific socio-demographic characteristics in the data set (S1 Table) to assure anonymity of our participants. The following information is not provided online: Age,

## Abstract

### Background

Being afraid of others' positive appraisal of oneself is called fear of positive evaluation. Fear of positive evaluation has been studied intensively in the context of social anxiety disorder (SAD). It is not known if individuals with borderline personality disorder (BPD) fear positive evaluation and which factors are associated with fear of positive evaluation in BPD.

### Methods

We applied the fear of positive evaluation scale and further self-report measures (e.g., social phobia inventory, rejection sensitivity questionnaire) to 36 patients with BPD, 29 patients with SAD and 35 healthy controls (HC).

### Results

A one-way ANOVA revealed that patients with BPD and patients with SAD reported significantly higher fear of positive evaluation than HC. Patients with BPD and SAD did not differ in their fear of positive evaluation. A hierarchical regression analysis revealed an association between rejection sensitivity and fear of positive evaluation in the BPD sample. However, this association disappeared when controlling for social anxiety.

### Conclusion

Our results indicate that individuals with BPD fear positive evaluation as much as individuals with SAD do, which has implications for clinical practice. Our results further imply that social anxiety is decisive for high fear of positive evaluation in patients with SAD and patients with BPD.

gender, medication intake, and information regarding co-morbid disorders.

**Funding:** We acknowledge support by the Open Access Publication Fund of the Freie Universität Berlin.

**Competing interests:** The authors have declared that no competing interests exist.

## Introduction

Borderline personality disorder (BPD) is a severe mental disorder that affects approximately 1.6% of the general population [1]. Individuals with BPD suffer from emotional instability, impulsive behavior, fear of abandonment, and strong social impairments [1, 2]. Researchers try to understand the nature of these strong social impairments to improve psychological interventions for BPD. There is extensive research showing that social impairments in BPD are associated with a negativity bias, which means that individuals with BPD process social information in a negative manner [3–7].

A new line of research revealed that this also applies to positive social information, in that way that individuals with BPD process and react differently to positive social information. In the context of these positivity impairments in BPD, it is important to study how individuals with BPD appraise positive social information. An interesting candidate to do so is fear of positive evaluation. This study examines fear of positive evaluation and its correlates in BPD in comparison to another clinical group and healthy individuals.

### Positivity impairments in BPD

Positivity impairments can be defined as alterations in the experience of positive affect as well as alterations in the processing of positive information [8–10]. Concerning the experience of positive affect, there is robust evidence that individuals with BPD experience less positive affect in their daily life (e.g., [11, 12]) and report to down-regulate positive affect (e.g., [13]). Moreover, individuals with BPD experienced positive affective states (e.g. to feel accepted, to feel safe) and cognitive states (e.g., "I trust myself") less frequently than individuals with another personality disorder [14]. Importantly, these impairments seem to predict recovery of BPD over time [15].

Concerning the processing of positive information, research revealed that individuals with BPD process positive information in a more negative manner. For example, individuals with BPD experienced less positive emotions such as pride and happiness after reading self-relevant appreciating sentences [16] and rated positive, self-relevant words as more negative than a non-clinical control group [17]. Moreover, individuals with BPD seem to be impaired in the processing of positive social feedback. An experimental study indicated that individuals with BPD integrate negative self-relevant social feedback to a greater extent than positive social feedback [18], while healthy individuals show the opposite updating bias (integrating positive feedback to a greater extent than negative). Another experimental study indicated that individuals with BPD change their social expectations in response to negative, but not positive, social feedback [19].

Hence, there is evidence for positivity impairments in BPD, which includes relevant findings for alterations in the processing of positive social feedback. However, it is not known if individuals with BPD appraise a positive social feedback/ evaluation differently.

### Fear of positive evaluation

Fear of evaluation is a hallmark feature of individuals with social anxiety disorder (SAD; [20]). Most research focused on fear of negative evaluation in SAD [1, 21]. However, recent research indicates that individuals with SAD are also characterized by fear of positive evaluation (e.g., [22, 23]). Fear of positive evaluation is defined as fearing others' favorable social appraisal [22].

The evolutionary model of social anxiety describes why social anxiety is characterized by fear of positive *and* negative evaluation [24]. According to this model, individuals in a group try to avoid a decrease, but also an increase in the social rank, as the latter might lead to

conflicts with more dominant group members. Consequently, fear of negative and positive evaluation is adaptive for social individuals as it decreases the likelihood of social conflicts.

Fear of positive evaluation has rarely been studied in other mental disorders than SAD and, to our knowledge, has not been studied in BPD. Reichenberger and Bleichert [25] asked for studies on fear of positive evaluation in BPD. They argued that fear of positive evaluation in BPD might be relevant, because high fear of positive evaluation [25] as well as BPD symptoms (e.g., [26]) are associated with social impairments.

## Correlates of fear of positive evaluation in BPD

There are preliminary results for an association between fear of positive evaluation and BPD symptoms. In an undergraduate sample, students with heightened symptoms of BPD reported high fear of positive evaluation [27]. However, this association disappeared when controlling for social anxiety. The author concluded that the association between BPD symptoms and fear of positive evaluation was due to high social anxiety in BPD [27].

Linehan [28] described a different approach for high fear of positive evaluation in BPD. She argued that individuals with BPD fear praise, because praise implies that the person will no longer require the support of others. In the therapeutic context, no requirement of further support could lead to the termination of sessions and the therapeutic relationship. Hence, individuals with BPD might fear positive evaluation because they fear abandonment/ rejection. This is especially interesting in the context of high rejection sensitivity in BPD (e.g., [29]).

## Research questions and hypotheses

In this study, we compared fear of positive evaluation in individuals with BPD to fear of positive evaluation in individuals with SAD and healthy controls (HC). Moreover, we examined the association of fear of positive evaluation, social anxiety and rejection sensitivity. Based on the results of Rodman [27] and the theoretical considerations of Linehan [28], the following hypotheses were examined:

1. We hypothesized that individuals with BPD show higher levels of fear of positive evaluation than healthy participants. On an exploratory level, we compared fear of positive evaluation in BPD to fear of positive evaluation in SAD.

2. We hypothesized that social anxiety explains most of the variance in fear of positive evaluation. Based on theoretical considerations [28], we assumed that specifically in individuals with BPD rejection sensitivity is associated with fear of positive evaluation over and above social anxiety.

## Materials and methods

The study was approved by the ethics committee of Freie Universität Berlin (ID 97 II /2016).

## Participants

Overall, 100 participants took part in the study: 35 HCs, 29 patients with SAD and 36 patients with BPD. A subsample of this sample was described in Weinbrecht et al. [30]. Table 1 displays sample characteristics as well as comorbid diagnoses of the BPD and SAD groups. Groups did not differ in age and gender (all $p > 0.53$).

**Table 1. Sample characteristics.**

|  | HC (*n* = 35) | SAD (*n* = 29) | BPD (*n* = 36) |
|---|---|---|---|
| Female, *n* (%) | 29 (83) | 23 (79) | 31 (86) |
| Age, *M (SD)* | 27.69 (5.66) | 28.83 (6.10) | 28.08 (4.95) |
| Number of comorbid diagnoses, *M (SD)* | 0 | 1.21 (1.15) | 1.70 (1.10) |
| Antidepressant medication, *n* (%) | 0 | 8 (27.59) | 13 (36.11) |
| MDE current, *n* (%) | 0 | 7 (24.18) | 2 (5.56) |
| MDE lifetime, *n* (%) | 0 | 8 (27.59) | 15 (41.67) |
| SAD, *n* (%) | 0 | 29 (100) | 1 (2.78) |
| Other anxiety disorders, *n* (%) | 0 | 0 | 11 (30.56) |
| PTSD, *n* (%) | 0 | 5 (17.24) | 11 (30.56) |
| BPD, *n* (%) | 0 | 0 | 36 (100) |
| AVPD, *n* (%) | 0 | 7 (24.18) | 0 |
| Other personality disorders, *n* (%) | 0 | 1 (3.45) | 3 (8.33) |

HC = Healthy Controls, SAD = Social Anxiety Disorder, BPD = Borderline Personality Disorder; MDE = Major Depressive Episode, PTSD = Posttraumatic Stress Disorder, AVPD = Avoidant Personality Disorder.

## Questionnaires

**Beck Depression Inventory–II (BDI-II; [31]).** We applied the German Version [32] of the BDI-II to measure severity of depressive symptoms. Participants have to rate the occurrence of depressive symptoms within the last two weeks on 21 items (range 0–63). The German version of the BDI-II shows good psychometric properties [33]. In our sample, Cronbach's α was excellent (α = 0.96).

**Brief Fear of Negative Evaluation Scale–Revised (BFNE-R; [21]).** We used the German version of the BFNE-R [34] to assess fear of negative evaluation by others. The German version of the BFNE-R contains 12 items (e.g., "I am frequently afraid of other people noticing my shortcomings") with a 5-point Likert scale ranging from 1 (*not at all characteristic of me*) to 5 (*extremely characteristic of me*). The final scores (sum of item scores) range from 12 to 60. The German version of the BFNE-R shows excellent psychometric properties [34]. In our sample, Cronbach's α was excellent (α = 0.95).

**Fear of Positive Evaluation Scale (FPES; [35]).** We used the German version of the FPES [36] to assess fear of positive evaluation by others. The FPES contains 10 items (e.g., "I don't like to be noticed when I am in public places, even if I feel as though I am being admired") with a 10-point Likert scale ranging from 0 (*not at all true*) to 9 (*very true*). Two of the 10 items are reversed-coded to detect response biases and were not used for the calculation of the total score (sum of item scores: range 0–72). The German version of the FPES shows good psychometric properties [36].

In our study, the FPES showed good psychometric properties. The internal consistency with Cronbach's α = 0.85 was good. To examine construct validity, we performed a confirmatory factor analysis examining if FPES and BFNE-R load on distinct factors. A root mean square residual (RMSR) of ≤ 0.08 indicates a good model fit. In our analysis, the model fit for the two-factor solution (FPES and BFNE-R are distinct factors) was good (RMSR = 0.05).

**Questionnaire of Thoughts and Feelings (QTF; [37]).** We applied the QTF [38] to measure BPD specific cognitions and emotions (range 1–5). The German version shows solid psychometric properties [38]. In our sample, Cronbach's α was excellent (α = 0.95).

**Rejection Sensitivity Questionnaire (RSQ; [39]).** We used the German short version of the RSQ to measure rejection sensitivity [29]. The RSQ-9 contains nine hypothetical

interpersonal situations, in which a significant other might refuse a request for support, guidance or companionship. Participants have to rate a) their expectation of being rejected (e.g., "I would expect that he or she would willingly agree to help me out.") and b) their anxiety of being rejected (e.g., "How concerned would you be over whether or not your friend would want to help you out?") on a 6-point Likert scale. The calculation of the total RSQ score is described in Gutz, Renneberg [40] and ranges from 1–36. The German version of the RSQ shows good psychometric properties [29]. In our sample, internal consistency was good (Cronbach's α = 0.89).

**Social Phobia Inventory (SPIN; [41]).** We applied the German Version [42] of the SPIN to measure severity of social anxiety symptoms (range 0–68). The SPIN consists of 17 items, with a 5-point Likert scale (from 0 = "*not at all true*" to 4 = "*extremely*"). The German version shows solid psychometric properties [42]. In our sample, Cronbach's α was excellent (α = 0.95).

## Procedure

Questionnaires were assessed online using the survey program Unipark (QuestBack GmbH, Germany). Participants were then invited to participate in an experimental study to assess EEG data on the processing of social participation in BPD and SAD (for results see [30]). At the lab, participants completed the BDI-II [31]. Moreover, if no diagnostic information was available, clinical psychologists conducted diagnostic interviews with the German versions [43] of SCID I and SCID II [44]. At the beginning and at the end of lab sessions, participants provided written informed consent.

We recruited participants via media advertisement, an inpatient clinic and two university outpatient departments. We only recruited participants between 18 and 40 years of age, who had no psychotic disorder, no current substance abuse/dependency and did not take psychotropic medication in the last 4 weeks.

## Statistical analysis

To compare fear of positive evaluation between groups, we performed a one-way ANOVA with *group* (3 levels: HC, SAD, BPD) as the independent variable and FPES scores as the dependent variable. Significant group effects were further examined with Bonferroni corrected post-hoc analyses. We used Hedges *g* as an effect size measure.

To examine the influence of rejection sensitivity (RSQ scores) and social anxiety (SPIN scores) on fear of positive evaluation (FPES scores), we performed a hierarchical regression analysis. First, we entered the group factor as a predictor to control for ecological fallacy. Next, we entered RSQ scores and then SPIN scores. In a last step, the interaction term between RSQ scores and group was entered. Assumptions (linearity, independent and normal distributed errors, homoscedasticity, multicollinearity) were not violated.

Analyses were conducted using R version 3.4.0 [45] and an alpha level of 0.05 was applied.

## Results

Fig 1 depicts box plots of group-specific FPES scores. Table 2 displays exact values (MD and SD) for all applied questionnaires. Individual questionnaire data are in the supporting information file (S1 Table).

We compared group-specific means in a one-way ANOVA to examine differences in fear of positive evaluation between patients with BPD, patients with SAD and HC. Groups differed significantly on FPES scores (see Table 2). The post-hoc analyses revealed that patients with BPD ($p < 0.001$, $g = 1.83$) and patients with SAD ($p < 0.001$, $g = 1.63$) reported higher FPES

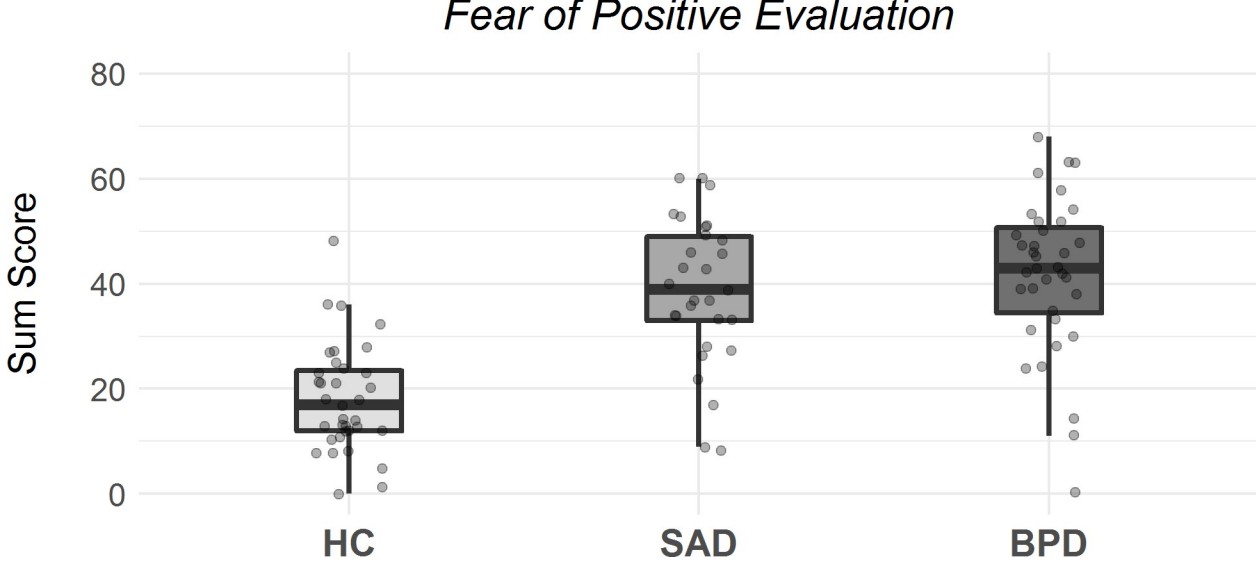

**Fig 1. Box plots for FPES scores with individual data points.** Boxes range from first to third quartile and represent the middle 50% of the data. Whiskers represent standard errors. HC = Healthy Controls, SAD = Social Anxiety Disorder, BPD = Borderline Personality Disorder.

scores than HC. Patients with SAD and patients with BPD did not differ significantly on their fear to be positively evaluated ($p = 1$, $g = 0.20$). In an exploratory analysis, we further compared differences between groups on the item level of the FPES. Clinical groups differed only on two items: Patients with BPD reported significantly higher values on two items related to being uncomfortable when receiving a compliment than patients with SAD (item 4 and 8, both $p < 0.01$).

Patients with BPD and patients with SAD reported higher rejection sensitivity than HC (both $p < 0.001$ and $g > 1.67$), but did not differ between each other ($p = 0.43$, $g = 0.32$). Comparable results were obtained for fear of negative evaluation (HC vs. SAD: $p < 0.001$, $g = 2.05$; HC vs. BPD: $p < 0.001$, $g = 1.24$; SAD vs. BPD: $p = 0.23$, $g = 0.45$). Patients with BPD reported the highest depressive symptoms (HC vs. BPD: $p < 0.001$, $g = 3.03$; SAD vs. BPD: $< 0.001$, $g = 1.35$; HC vs. SAD: $p < 0.001$, $g = 1.54$).

**Table 2. Results of self-report questionnaires.**

| | HC | SAD | BPD | ANOVA |
|---|---|---|---|---|
| | *M (SD)* | *M (SD)* | *M (SD)* | *F (2, 97)* |
| Fear of Positive Evaluation [FPES] | 18.06 (10.30) | 38.69 (13.99) | 41.67 (14.80) | 33.12* |
| Fear of Negative Evaluation [BFNE-R] | 29.91 (10.50) | 48.62 (6.74) | 44.06 (12.01) | 30.3* |
| Rejection Sensitivity [RSQ] | 8.43 (3.24) | 15.97 (5.59) | 17.93 (6.54) | 31.37* |
| Social Anxiety [SPIN] | 12.86 (8.41) | 41.79 (11.47) | 37.03 (12.91) | 65.45* |
| BPD Specific Cognitions [QTF] | 1.43 (0.45) | 2.32 (0.77) | 3.53 (0.81) | 81.97* |
| Depressive Symptoms [BDI-II] | 4.10 (5.10) | 16.86 (10.78) | 32.51 (12.01) | 75.95* |

HC = Healthy Controls, SAD = Social Anxiety Disorder, BPD = Borderline Personality Disorder; FPES = Fear of Positive Evaluation Scale, BFNE-R = Brief Fear of Negative Evaluation Scale–Revised, SPIN = Social Phobia Inventory, BDI-II = Beck Depression Inventory II, QTF = Questionnaire of Thoughts and Feelings, RSQ = Rejection Sensitivity Questionnaire.

* $p < 0.001$.

QTF scores reflected the diagnostic grouping: Patients with BPD reported more BPD specific cognitions than patients with SAD ($p < 0.001$, $g = 1.50$) and HC ($p < 0.001$, $g = 3.14$). Patients with SAD reported more BPD specific cognitions than HC ($p < 0.001$, $g = 1.44$). SPIN scores did not entirely reflect the diagnostic grouping: Patients with SAD reported higher social anxiety than HC ($p < 0.001$, $g = 2.88$), but did not differ from patients with BPD (SAD vs. BPD: $p = 0.26$, $g = 0.38$; HC vs. BPD: $p < 0.001$, $g = 2.19$).

## Association between social anxiety, rejection sensitivity and fear of positive evaluation

In a further step, we looked at correlates of fear of positive evaluation in BPD. FPES scores were highly correlated with the other questionnaires (see Table 3). Interestingly, fear of positive evaluation and social anxiety were significantly correlated in all groups. However, fear of positive evaluation and rejection sensitivity were only significantly correlated in patients with BPD (see Table 3).

The hierarchical regression analysis (see Table 4) revealed that rejection sensitivity explained 48.37% of the variance in fear of positive evaluation while controlling for diagnostic group, $F(1,96) = 26.02$, $p < 0.001$, adj. $\Delta R^2 = 0.09$ (see Model 2, Table 4). Adding social anxiety scores to the model significantly increased the explained variance in fear of positive evaluation, $F(1,95) = 44.07$, $p < 0.001$, adj. $\Delta R^2 = 0.16$. In this model, only social anxiety significantly predicted fear of positive evaluation while controlling for diagnostic group (see $b$-values of Model 3, Table 4). Adding the interaction term did not explain significantly more variance in fear of positive evaluation, $F(2,93) = 1.16$, $p = 0.32$, adjusted $\Delta R^2 = 0.01$.

## Discussion

To our knowledge, this is the first study that examined fear of positive evaluation in individuals with BPD. Individuals with BPD did not differ in their fear to be positively evaluated from individuals with SAD.

Furthermore, we examined which factors were associated with high fear of positive evaluation in a regression analysis. As hypothesized, social anxiety explained most of the variance in the fear to be positively evaluated.

**Table 3. Correlations between FPES and other questionnaires.**

|  | HC ($n = 35$) | SAD ($n = 29$) | BPD ($n = 36$) |
|---|---|---|---|
|  | *FPES r* | *FPES r* | *FPES r* |
| Fear of Negative Evaluation [BFNE-R] | 0.50* | 0.10 | 0.68** |
| Rejection Sensitivity [RSQ] | 0.09 | 0.33 | 0.55** |
| Social Anxiety [SPIN] | 0.55** | 0.44* | 0.80** |
| BPD Specific Cognitions [QTF] | 0.37* | 0.23 | 0.60** |
| Depressive Symptoms [BDI-II] | 0.28 | 0.29 | 0.44* |

HC = Healthy Controls, SAD = Social Anxiety Disorder, BPD = Borderline Personality Disorder; FPES = Fear of Positive Evaluation Scale, BFNE-R = Brief Fear of Negative Evaluation Scale–Revised, SPIN = Social Phobia Inventory, BDI-II = Beck Depression Inventory II, QTF = Questionnaire of Thoughts and Feelings, RSQ = Rejection Sensitivity Questionnaire.

* $p < 0.05$

** $p < 0.005$.

**Table 4. Hierarchical regression analyses predicting fear of positive evaluation.**

|  | *F*-statistic | adj. $R^2$ | *b* | SE *b* | *β* |
|---|---|---|---|---|---|
| Model 1: *M* (SAD) = 38.69 | $F(2, 97) = 33.12$ | 0.39** |  |  |  |
| SAD vs. HC |  |  | -20.63** | 3.30 | -1.22** |
| SAD vs. BPD |  |  | 2.99 | 3.28 | 0.18 |
| Model 2: *M* (SAD) = 22.98[a] | $F(3, 96) = 31.91$ | 0.48** |  |  |  |
| SAD vs. HC |  |  | -13.21** | 3.51 | -0.78** |
| SAD vs. BPD |  |  | 1.06 | 3.06 | 0.06 |
| Rejection Sensitivity [RSQ] |  |  | 0.98** | 0.23 | 0.39** |
| Model 3: *M* (SAD) = 4.18[b] | $F(4, 95) = 45.61$ | 0.64** |  |  |  |
| SAD vs. HC |  |  | 1.85 | 3.70 | 0.11 |
| SAD vs. BPD |  |  | 5.39 | 2.63 | 0.32 |
| Rejection Sensitivity [RSQ] |  |  | 0.40 | 0.21 | 0.16 |
| Social Anxiety [SPIN] |  |  | 0.67** | 0.10 | 0.67** |

HC = Healthy Controls, SAD = Social Anxiety Disorder, BPD = Borderline Personality Disorder; SPIN = Social Phobia Inventory, RSQ = Rejection Sensitivity Questionnaire.

[a] group mean of SAD sample on FPES when controlling for RSQ scores

[b] group mean of SAD sample on FPES when controlling for RSQ and SPIN scores

* $p < 0.05$

** $p < 0.005$.

## High fear of positive evaluation in individuals with BPD

Until now, fear of positive evaluation in clinical samples seemed to be highest in SAD [25]. For example, fear of positive evaluation was higher in individuals with SAD compared to individuals with other anxiety disorders [46]. In non-clinical samples, fear of positive evaluation was more strongly related to social anxiety than to depressive symptoms [23, 35]. Our results show that individuals with SAD do not differ from individuals with BPD in their fear of positive evaluation.

Fear of positive evaluation has been described in the context of positivity impairments in SAD (see [8, 9] for reviews). Our finding on high fear of positive evaluation in BPD adds to literature on positivity impairments in BPD. For example, individuals with BPD seem to have problems integrating positive (self-referential) social information (e.g., [18, 19, 30, 47]). Our study extends previous findings showing that individuals with BPD also appraise positive social information more anxiously than healthy individuals do.

To shed more light on high fear of positive evaluation in BPD, we compared differences between clinical groups on the item level of FPES in an exploratory analysis. Individuals with BPD reported to be more uncomfortable when receiving a compliment than individuals with SAD. It could be speculated that a compliment is incongruent with the negative self concept (e.g., [48]) of individuals with BPD. Therefore, a compliment might trigger unwanted negative emotions such as anger (e.g., "the therapist is lying to me") or shame (e.g., "the therapist has no idea how unworthy I am").

What are the implications of high fear of positive evaluation in BPD? High fear of positive evaluation has been associated with social impairments and less quality of life (see [25] for a review). Fear of positive evaluation might contribute to the well described long term impaired psychosocial functioning in BPD [26]. Moreover, fear of positive evaluation has been associated with diminished positive affect (see [25] for a review). Diminished positive affect was also found in BPD [14] and has been related to higher BPD symptom severity [12]. Future research

should examine if fear of positive evaluation contributes to diminished positive affect as well as social impairments in BPD.

What are the clinical implications of high fear of positive evaluation in BPD? Therapists should be aware that complimenting or giving positive feedback might be frightening and/ or difficult to accept for patients with BPD. Therefore, it could be helpful if the therapist prefaces compliments to the patient by pointing out that this might trigger aversive emotions. More-over, therapist and patient should explore why a compliment triggers negative emotions (e.g., the compliment is schema-incongruent, the patient is suspicious regarding the intentions behind the compliment). This way, the patient might learn to understand the experience of negative emotions before or after receiving a positive social feedback. In the long-term and accompanied by further interventions (e.g., development of a suitable skill to accept positive feedback), this might help patients with BPD to experience less negative and more positive emotions in the context of a positive social feedback.

## Association between social anxiety, rejection sensitivity and fear of positive evaluation

We also examined which factors are associated with high fear of positive evaluation. In line with our assumption, only in patients with BPD, rejection sensitivity was associated with fear of positive evaluation. However, the regression analysis revealed that the association between rejection sensitivity and fear of positive evaluation disappeared when controlled for social anx-iety and that social anxiety accounts for most of the variance in fear of positive evaluation. Hence, social anxiety might have driven the association between rejection sensitivity and posi-tive evaluation in our sample. This is in line with previous results showing that the association between BPD symptoms and fear of positive evaluation was driven by high social anxiety in BPD [27].

However, this finding is limited by the fact that social anxiety and fear of positive evaluation were highly correlated in our BPD sample ($r = 0.80$), which raises the question if the applied questionnaires measure the same underlying construct. Indeed, there is an overlap in some items of both applied questionnaires (e.g., SPIN: "I am afraid of people in authority", "I avoid activities in which I am the center of attention."; FPES: "I feel uneasy when I receive praise from authority figures.", "I don't like to be noticed when I am in public places, even if I feel as though I am being admired."). However, the high correlation between both questionnaires was specific for the BPD sample. In healthy participants and the SAD sample, the correlation was lower ($r = 0.55$, $r = 0.44$) and comparable to previous studies [23, 46]. Moreover, fear of positive evaluation relates to being afraid of others' positive appraisal, while social anxiety is characterized by fear of negative evaluation and less well researched fear of positive evaluation (e.g., [23, 49]). This favors the distinctiveness of both constructs.

It is noteworthy that in the BPD sample, different self-report measures were highly corre-lated. It is possible that a generalized negative affectivity or a negativity bias (3–6) drives this correlation pattern.

It should further be noted that individuals with SAD did not differ in their rejection sensi-tivity from individuals with BPD. This is in contrast to previous findings that individuals with BPD are characterized by higher rejection sensitivity than individuals with SAD [29, 40]. A possible explanation for this is the high symptom load of participants with SAD in our study (see [30]).

Future studies need to clarify the underlying mechanism of high fear of positive evaluation in BPD. As speculated above, a positive evaluation might be incongruent with the negative self concept (e.g., [48]) of individuals with BPD or related to impairments in social cognition in

BPD (e.g., [3, 50]), which lead to being suspicious regarding the intentions behind a positive evaluation.

## Strength and limitations

This is the first study that compared fear of positive evaluation in BPD to fear of positive evaluation in SAD and healthy individuals. Data allowed us to examine if high fear of positive evaluation is specific to individuals with SAD or if it is also present in BPD. Diagnostic groups were confirmed with structured clinical interviews and were reflected in BPD specific cognitions and emotions (QTF scores; [38]). Moreover, we were able to confirm that the FPES is a valid and reliable questionnaire [36].

The following limitations need to be considered: 1) we relied exclusively on self-report measures, 2) we designed a cross-sectional study, which provided no conclusion on causality, and 3) we were not able to look at sex differences in fear of positive evaluation.

## Conclusion

This study showed that individuals with BPD highly fear positive evaluation. This is important for clinical practice. Therapists should be aware that complimenting or giving positive feedback might be frightening to patients with BPD and apply suitable interventions.

Future research should examine why individuals with BPD fear positive evaluation and explore strategies to diminish this fear of positive evaluation.

## Supporting information

**S1 Table. Complete data set of questionnaires.**
(XLSX)

## Acknowledgments

We thank Lars Schulze, Lydia Fehm, Jana Zitzmann, Konstantin Nikolaidis, and Marilú Nolte for their help with gathering the data and with the implementation of the study.

## Author Contributions

**Conceptualization:** Anna Weinbrecht, Stefan Roepke, Babette Renneberg.

**Data curation:** Anna Weinbrecht, Stefan Roepke.

**Formal analysis:** Anna Weinbrecht.

**Methodology:** Anna Weinbrecht, Babette Renneberg.

**Project administration:** Anna Weinbrecht.

**Resources:** Stefan Roepke, Babette Renneberg.

**Supervision:** Stefan Roepke, Babette Renneberg.

**Writing – original draft:** Anna Weinbrecht.

**Writing – review & editing:** Stefan Roepke, Babette Renneberg.

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
