## [Decision Letter · Decision Letter 0]

18 Jun 2020

PONE-D-20-12363

Fear of positive evaluation in borderline personality disorder

PLOS ONE

Dear Dr. Weinbrecht,

Thank you for submitting your manuscript to PLOS ONE. After careful consideration, we feel that it has merit but does not fully meet PLOS ONE’s publication criteria as it currently stands. Therefore, we invite you to submit a revised version of the manuscript that addresses the points raised during the review process.

We look forward to receiving your revised manuscript.

Kind regards,

Svenja Taubner

Academic Editor

PLOS ONE

Journal Requirements:

We thank Lars Schulze, Lydia Fehm, Jana Zitzmann, Konstantin Nikolaidis, and Marilú

Nolte for their help with gathering the data and with the implementation of the study.

We acknowledge support by the Open Access Publication Fund of the Freie Universität

Berlin.

The authors received no specific funding for this work.

Reviewers' comments:

Reviewer's Responses to Questions

**Comments to the Author**

1. Is the manuscript technically sound, and do the data support the conclusions?

Reviewer #1: No

Reviewer #2: Yes

2. Has the statistical analysis been performed appropriately and rigorously? 

Reviewer #1: No

Reviewer #2: Yes

3. Have the authors made all data underlying the findings in their manuscript fully available?

Reviewer #1: Yes

Reviewer #2: Yes

4. Is the manuscript presented in an intelligible fashion and written in standard English?

Reviewer #1: No

Reviewer #2: Yes

5. Review Comments to the Author

Reviewer #1: This manuscript examines the role of fear of positive appraisal and rejection sensitivity in borderline personality disorder (BPD) comparing three groups: healthy controls, patients with BPD, and patients with social anxiety disorder. While this paper has the potential to be a contribution to the growing literature about the role of difficulty with positive appraisal and BPD, in its current form is underdeveloped in its literature review as well as its explanation of findings.

My first major criticism of this draft is that the introduction suggests that there are distinct features of BPD samples that would suggest fear of positive appraisal, but the findings and implications of these studies could be better explained. This literature is summarized by the authors using 4 papers which are swiftly described in jargon laden terms which makes it difficult to know the actual studies or the relevant take home points. The fifth study, reference 13 is cited to suggest there is a "positivity impairment" in BPD, based on a chapter on social anxiety disorder. More can be clarified about what this positivity impairment involves, and ignores literature of BPD and positive affects and cognition (e.g. Reed & Zanarini, 2011; Reed, Fitzmaurice, and Zanarini 2012). There is no discussion of the existing literature on positive affect and BPD, which is confusing given the literature background presented is sparse and brief. The authors then turn to the role of fear of positive evaluation in social anxiety, using an evolutionary model, but do not elaborate in any way how this would relate to BPD. This paper would be a better contribution if authors gave more space for summarizing more clearly this interesting literature.

My second criticism, is that the study hypotheses are nonspecific in terms of what was to be found in the comparison between social anxiety and BPD. The finding that rejection sensitivity and fear of positive appraisal did not differentiate the groups could be framed in a more specific way and is not clearly explained. Given the lack of overlap between the SAD and BPD groups it is hard to determine how social anxiety drives fear of positive evaluation for the entire sample or how it washes out the effects of rejection sensitivity. The authors do not explain this at all in the conclusions and only reference this study's similarity to a report in an unpublished dissertation. A fuller exploration for why occurred would help the manuscript.

My last criticism is that the clinical implication of these findings are mentioned but not explicit what would be done clinically in terms of intervention or management. The authors say a therapist should “apply suitable interventions” after exploring the emotional reaction of their patient. Could they specify some examples of suitable interventions?

A number of copy editing problems are as follows: Pg. 4, second full sentence, should read “adaptive”, instead of “adaptable”; Pg. 12 last paragraph, should read “explained”, instead of “exlained”, line 58 says letter instead of latter) was: Line 151 – they mention a measure (LPS-4) that isn’t spelled out nor described in their measures. Seems to be a measure of IQ though, so not a main outcome measure, not sure if it needs more description.

Reviewer #2: This study investigates fear of positive evaluation in BPD patients as compared to SAD and HC. It’s a well-structured, well-written paper and I only have some minor comments and suggestions for additional discussion.

In the intro, the authors seem to make the argument that fear of positive evaluation in BPD patients may have a different nature than in SAD and be more determined by BPD patient’s fear of rejection, following Linehan’s theory. One would assume the authors would expect that rejection sensitivity might have additional explanatory value beyond social anxiety for BPD patients, based upon their intro (and their statement that they base their hypothesis on Linehan’s theory). I wonder why they haven’t put this hypothesis.

Some more information may be provided on the nature of the SAD sample and whether there are indications for comorbidity with PDs in the SAD sample. On the other hand, some information on the presence of SAD and other anxiety disorders in the BPD sample. This may help to clarify how different both samples are and what kind of SAD vs BPD sample these are.

There is a very high correlation in the BPD sample between social anxiety and fear of positive evaluation of .80. At least in this sample measures of social anxiety and fear of positive evaluation seem to be measuring almost the same construct. I think this needs extra attention from the authors as this may reflect a problem with their measures.

Remarkably in the BPD sample is the general rather high level of correlations between all variables and fear of positive evaluations (and maybe also among each other) as compared to the other samples showing a more diverse pattern of correlations. Could the authors discuss this difference? Could it be related to a generalized negative affectivity that characterizes BPD?

The authors discuss very briefly implications and I would like them to add a bit to this discussion given the prominent view in many (behavioral) approaches to reinforce patients by making compliments and to assist parents in complimenting their (BPD) kids. These findings seem to suggest that such an approach only triggers negative feelings in BPD patients. How do these findings fit with therapeutic recommendations?

The authors may discuss their findings also in relation to other possible explanations in the literature with regard to fear for positive evaluation in BPD patients, that may be targeted in future studies, e.g. problems in social cognition making BPD patients suspicious regarding the intentions behind positive feedback; or the discrepancy between positive evaluations and their own low self-esteem.

6. PLOS authors have the option to publish the peer review history of their article (what does this mean?). If published, this will include your full peer review and any attached files.

Reviewer #1: No

Reviewer #2: No

---

## [Author Response · Author response to Decision Letter 0]

17 Jul 2020

We highly appreciate your feedback on our initial version, and have done our best to address all of the comments/suggestions provided. 

We provide a detailed discussion of all corrections in the document "Response to Reviewers".

---

## [Decision Letter · Decision Letter 1]

6 Aug 2020

Fear of positive evaluation in borderline personality disorder

PONE-D-20-12363R1

Dear Dr. Weinbrecht,

We’re pleased to inform you that your manuscript has been judged scientifically suitable for publication and will be formally accepted for publication once it meets all outstanding technical requirements.

Kind regards,

Svenja Taubner

Academic Editor

PLOS ONE

Additional Editor Comments (optional):

Reviewers' comments:

Reviewer's Responses to Questions

**Comments to the Author**

1. If the authors have adequately addressed your comments raised in a previous round of review and you feel that this manuscript is now acceptable for publication, you may indicate that here to bypass the “Comments to the Author” section, enter your conflict of interest statement in the “Confidential to Editor” section, and submit your "Accept" recommendation.

Reviewer #2: All comments have been addressed

2. Is the manuscript technically sound, and do the data support the conclusions?

Reviewer #2: Yes

3. Has the statistical analysis been performed appropriately and rigorously? 

Reviewer #2: Yes

4. Have the authors made all data underlying the findings in their manuscript fully available?

Reviewer #2: Yes

5. Is the manuscript presented in an intelligible fashion and written in standard English?

Reviewer #2: Yes

6. Review Comments to the Author

Reviewer #2: In would like to thank the authors for having addressed the issues raised and thereby providing somewhat more context for their findings.

7. PLOS authors have the option to publish the peer review history of their article (what does this mean?). If published, this will include your full peer review and any attached files.

Reviewer #2: No

---

## [Editor Report · Acceptance letter]

10 Aug 2020

PONE-D-20-12363R1 

Fear of positive evaluation in borderline personality disorder 

Dear Dr. Weinbrecht:

I'm pleased to inform you that your manuscript has been deemed suitable for publication in PLOS ONE. Congratulations! Your manuscript is now with our production department. 

Kind regards, 

on behalf of

Dr. Svenja Taubner 

Academic Editor

PLOS ONE